# A Residual UNet Denoising Network Based on Multi-Scale Feature Extraction and Attention-Guided Filter

**DOI:** 10.3390/s23167044

**Published:** 2023-08-09

**Authors:** Hualin Liu, Zhe Li, Shijie Lin, Libo Cheng

**Affiliations:** 1School of Mathematics and Statistics, Changchun University of Science and Technology, Changchun 130022, China; 2021100024@mails.cust.edu.cn (H.L.); 2020000097@mails.cust.edu.cn (S.L.); chenglb@cust.edu.cn (L.C.); 2Laboratory of Remote Sensing Technology and Big Data Analysis, Zhongshan Research Institute, Changchun University of Science and Technology, Zhongshan 528437, China

**Keywords:** image denoising, multi-scale feature, attention-guided filter, UNet, residual network

## Abstract

In order to obtain high-quality images, it is very important to remove noise effectively and retain image details reasonably. In this paper, we propose a residual UNet denoising network that adds the attention-guided filter and multi-scale feature extraction blocks. We design a multi-scale feature extraction block as the input block to expand the receiving domain and extract more useful features. We also develop the attention-guided filter block to hold the edge information. Further, we use the global residual network strategy to model residual noise instead of directly modeling clean images. Experimental results show our proposed network performs favorably against several state-of-the-art models. Our proposed model can not only suppress the noise more effectively, but also improve the sharpness of the image.

## 1. Introduction

As a typical problem in the field of low-level image processing, image denoising aims to recover the clean image from the observed image that is corrupted by some noise. Since both the clean image and the noise are unknown, it is critical to remove the noise while preserving the details of the image.

Traditional image denoising approaches, including filtering-based methods and model-based methods, have always been the main image denoising methods. For example, non-local mean (NLM) [1], block matching and three-dimensional filtering (BM3D) [2], wavelet transform [3,4,5], have become the most advanced image denoising methods. However, these traditional denoising methods tend to blur the image texture and reduce the image visual quality.

Compared with traditional denoising methods, convolutional neural networks (CNNs) have achieved state-of-the-art performances in some representative image denoising tasks. The popularity of CNNs in image denoising can be explained from two aspects. On the one hand, CNN has the advantages of speed, performance, and generalization ability compared with traditional noise removal devices [6,7]. On the other hand, recent research has shown that CNN-based methods can be inserted into model-based optimization methods to solve more complex image restoration tasks [8,9], which can promote the application ability of the CNN models.

By constructing a learnable deep neural network, deep learning denoising methods learn the mapping from damaged images to clean images. Moreover, some classical module methods were inserted to CNNs in order to improve the efficiency of the denoising task efficiently. Feng et al. developed a trainable nonlinear reaction diffusion (TNRD) model to remove the Poisson noise quickly and accurately [10]. The feed-forward denoising convolutional neural networks (DnCNNs) [7] focused on the complementary role of the residual learning and batch normalization in image restoration. Memory blocks were introduced in a persistent memory network (MemNet) for image recovery [11]. The popular attention mechanism was used in an attention-guided denoising convolutional neural network (ADNet) [12]. A convolutional blind denoising network (CBDNet) [13] contains a noise estimation subnet with asymmetric learning to realize the noise level estimation.

Although deep learning-based denoising methods have achieved an excellent denoising effect, it is very difficult to further improve the performance of deep learning-based denoising methods, because they need to increase the depth or width of the network, which will encounter a sharp increase in training parameters. Most deep denoising networks lack the adaptability since these models need to be trained for each noise level, which may induce a poor performance for other noise levels. Therefore, it is worth investigating how to explore and utilize the existing denoising networks, so as to achieve more effective denoising. Multi-scale features are very useful in the field of image quality and visual saliency in computer vision. For example, Varge [14] introduced a no-reference image quality assessment with multi-scale orderless pooling of deep features and Li et al. [15] proposed a visual saliency approach based on multiscale deep features. Beyond that, some enhancement techniques are feasible to improve the recovery quality of deep learning-based denoising methods, such as Tukey’s shrinking strategy [16] and the multi-level wavelet transform [17].

In this paper, we propose a residual-dense neural network (MAGUNet) that incorporates multi-scale feature extraction blocks and attention-guided filter blocks for image denoising. Our proposed model has the ability to compete with the latest denoising methods. Based on the UNet architecture, our model consists of a shrunk subnet and an extended subnet. In the shrink subnetwork, the input block is constructed by the void convolution and multi-scale feature extraction block to extract more useful features from the input noise image. An attention-guided filter block is introduced to restore the image information after each down-sampling operation. Although our model has a larger number of parameters than the conventional methods, our model allows for a smaller number of multiplication operations compared with the models with similar or even higher complexity. Massive experiments have shown that our MAGUNet model outperforms the most advanced denoising methods such as BM3D, FFDNet, RDUNet, and MSAUNet. The contributions of this paper are summarized as follows:This paper proposes the MAGUNet model, which extracts more useful features by expanding the acceptance domain, so that our model can achieve a better balance between efficiency and performance.The attention-guided filter block is designed to retain the details of the image information after each down-sampling operation.The experiment results demonstrate the superiority of the MAGUNet model against the competing methods.

The remainder of this paper is organized as follows. The related work is reviewed in Section 2. Section 3 provides our proposed MAGUNet model. The experimental results are provided in Section 4. Section 5 presents some related discussion. The conclusion is given in Section 6.

## 2. Related Work

### 2.1. CNNs for Image Denoising

At present, there are many methods based on the neural networks to deal with the image denoising problem. Jain and Seung [18] proposed the earliest CNN model for natural image denoising. Burger et al. [19] proposed a multi-layer perceptron (MLP) block that allows the neural network to achieve better results than the BM3D [2] method. Zhang et al. [7] proposed a deep convolutional neural network (DnCNN) for image denoising, which improves the denoising performance by stacking multiple convolution layer blocks.

It is effective to use skip connection operation to enhance the expressiveness of CNN denoising models. By integrating the short-term memory and long-term memory, Tai et al. [11] proposed a deep end-to-end persistent memory network for image restoration, which enhances the influence of the shallow layer through recursion. The fast and flexible denoising CNN(FFDNet) [20] introduced noise feature graphs that deal with non-uniform noise levels to reduce the sampled subimages. A generative adversarial neural network (GAN) is proposed to estimate noise distribution and generate noise samples [21].

Recent models, such as DHDN [22], DIDN [23], and RDN [24], have improved the baseline results established by DnCNN and FFDNet models. However, these models significantly increase the number of parameters to achieve the improvement. Presently, UNet model [25] is one of the most widely used in the autoencoder architectures. Liu et al. [17] proposed the MWCNN model, which combines the wavelet transform and convolutional layer in the UNet model instead of simple convolution and maximum polarization. Wang et al. [26] extended the MWCNN model by adding residual-dense blocks to each layer of the model. The DIDN model proposed by Yu et al. [22] utilized several U-Net-based blocks, which may change the image size many times. He et al. [27] proposed the residual U-Net network (ResNets) to solve the problem about network degradation with the increase of the network depth. Dense networks [23] reuse each generated feature map to subsequent convolution within the same convolution block. Additionally, several image denoising methods based on residual learning and dense connectivity have been proposed. Zhang et al. [28] proposed a depth residual network with a linear element layer of parametric rectification for image recovery tasks. A residual dense neural network (RDUNet) for image denoising [29] combines dense concatenated convolutional blocks by feature graphs in the encoding and decoding parts.

### 2.2. Multi-Scale Feature Extraction

To capture more contextual information in CNNs, increasing the acceptance field size is a common technique. However, this usually requires expanding the depth and width of the network, which produces more parameters in the model. An alternative approach is to use dilated convolution, which can extract multi-scale information while keeping the feature map size constant. The dilated convolution is particularly useful for detection and segmentation tasks, as it can detect large targets and accurately locate them. Shallow layers of neural networks tend to have smaller acceptance domains, but they can learn and transmit image details to deeper layers for feature integration. As the network deepens, there may be a lack of long-term dependencies between features, which can be addressed by broadening the network and extracting richer features [30]. Different networks, such as GoogleNet [31] and CFBI+ [32], have used multi-scale approaches to enhance the expression ability and robustness of the neural network framework. The multi-scale adaptive network proposed by Gou et al. [33] integrates the intra-scale features and cross-scale complementarity into the network design at the same time. Zou et al. designed a dual attention to adaptively reinforce important channels and spatial regions [34]. Li et al. proposed a multi-scale feature fusion network for CT image denoising by combining multiple feature extraction modules [35]. In the multi-scale feature extraction blocks of our proposed MAGUNet model, the residual dilated convolution blocks were utilized to balance the number of parameters and the performance of feature extraction.

### 2.3. Attentional Mechanisms

As we all know, it is a great challenge to extract suitable features from a given complex background image. Zhu et al. [36] proposed an attention mechanism that combines the training flow and tracking task in a deep learning framework. Karri et al. [37] presented an interpretable multi-module semantically guided attention network including the location attention, channel-wise attention, and edge attention modules, so as to extract the most important spatial-, channel-, and boundary-related features. Fan et al. [38] proposed a new attention ConvNeXt model by introducing a parametrically free attention block. Yan et al. [39] proposed an attention-guided dynamic multi-branch neural network to obtain high-quality underwater images. Wang et al. [40] introduced an attention mechanism into the discriminator to reduce excessive attention and retain more detailed feature information. As an anisotropic filter, a guided filter [41] can preserve edge details efficiently. Wu et al. [42] proposed deep guided filter networks to deal with the problem about limited joint up-sampling capability. In light of the structure of the guide image, Ying et al. [43] generated a set of guide filters to preserve the edge smoothing. By exploring the fusion of the attention mechanism and guided filter, we propose the attention-guided filter block based on the ability of learning important features.

## 3. Methods

### 3.1. Network Structure

The architecture of our proposed network is shown in Figure 1, which is mainly composed of a multi-scale feature extraction block (MFE), an attention-guided filter block (AGB), and a residual denoising block (RDB).

In this paper, we deal with the noisy image y destroyed by the additive white Gaussian noise n. The image denoising problem can be formulated as finding the argument function F(⋅,θ) on the trainable parameter vector θ such that the estimated clean image $\hat{x}$ is computed by
(1)$$\hat{x}=F(y,\theta )$$

Since the damaged image contains most of the structure of the clean image, it is reasonable to retain the structure information by estimating the noise. To that end, we assume that there exists a parameter mapping H(⋅,θ) such that H(y,θ)≈−n. Therefore, the denoising parameter model based on residual learning can be written as follows:(2)F(y,θ)=H(y,θ)+y

Let $\left\{x_{i},\right.{\left.y_{i}\right\}}_{i=1}^{N}$ be the training dataset, where $y_{i}$ is a noisy image and $x_{i}$ is the corresponding clean image. For a given balancing factor λ, the parameter is computed by solving the following optimization problem:(3)$$\theta^{*}=\underset{\theta }{\arg\min}\frac{1}{N}\sum_{i=1}^{N}{\left\|F(y_{i};\theta )-x_{i}\right\|}_{1}+\frac{\lambda }{2}{\left\|\theta \right\|}^{2}$$
where the first and second terms are, respectively, the fidelity term and the regularization term.

From Figure 1, we can find that the main architecture of the mapping F(⋅,θ) is associated with the MAGUNet model of mapping H(⋅,θ). The main body of MAGUNet is composed of the encoder and decoder structures. The encoder part is responsible for extracting the low-level features of the image. The decoder part is responsible for recovering high-level features of the image while removing noise. The encoder and decode parts are connected by a series of residual modules. On the basis of the UNet, our MAGUNet model introduces the multi-scale feature extraction blocks as input blocks and attention-guided filter blocks after down-sampling.

In the encoding phase, there is one denoising block and one attention-guided filter block in each layer, and there are two denoising blocks in the decoding phase. The output of each coding layer adopts the down-sampling by means of a convolution kernel of size 2 × 2 and a step of size 2. Each step of down-sampling will double the number of feature maps to reduce the loss of information from one level to another. The up-sampling of the decoding layer is performed by the transposing convolution. The feature fusion between the coding layer and decoding layer is carried out by a skip connection. After each operation of up-sampling and with the skip connection, our model executes the 3 × 3 convolution to reduce the number of features and smooth the up-sampled features, while preserving the most important information of the source image. Our MAGUNet model mostly uses the PReLU activation functions. When the PReLU function is adopted, the number of trainable parameters is the same as the number of feature maps in the corresponding layer, which implies the PReLU function can improve the flexibility of the model without introducing a large number of additional parameters. The sigmoid and GELU activation functions are used only in the attention-guided filter layer. Because the nonlinearity and dropout jointly determine the output of neurons, the use of GELU activation function can make the probability of neuron output higher and reduce over-fitting. The input block used for multi-scale feature extraction is shown in Figure 2. Spatial information can be reasonably used to predict the actual values of the given pixels for the denoising task. In the local area, the current pixel of the predicted image and its adjacent pixels have similar pixel values. A high noise level usually requires a larger patch size to capture the contextual information. One way to obtain more spatial information is to select the convolution kernel of size larger than 3 × 3, which may increase the number of parameters in the spatial dimension. Our MAGUNet model generates 64 feature maps by 3 × 3 and 7 × 7 convolution kernels, so as to increase the receptive field and connect different features. The output block reduces the number of feature maps through two Conv 3 × 3 + PReLU operations to match the size of the input noise image and produce an estimate of residual noise. The corresponding output is used for global residual learning, which adds the corresponding result to the input image, so as to attain the denoised image.

### 3.2. Main Structure Module

#### 3.2.1. Multi-Scale Feature Extraction Block

Although the down-sampling can increase the receptive field, the spatial resolution is reduced. The dilated convolution can be used to enlarge the receptive field without decreasing resolution. The dilated convolution has a parameter called the dilation rate, which controls the accepted area of the convolution core. Hence, when the different dilation rates are set, the receptive fields will be different and multi-scale information will be acquired. Indeed, the dilated convolution can arbitrarily expand the receptive field without introducing additional parameters.

Moreover, in order to skip some layers that do not increase the overall accuracy value, the skip connection is added to every two layers. The corresponding architecture is shown in Figure 2a. The specific steps of residual dilated convolution block are as follows. First, we perform the 3 × 3 dilated convolution and PReLU operations on the input image, where the corresponding parameters are padding = 1, dilation = 1. After concatenating the input image, we perform the 3 × 3 dilated convolution and PReLU operations on the concatenated image, where padding = 2 and dilation = 2. Next, we perform convolution and PReLU operations to recover the channel number of the input image. Finally, we add the original input image to attain the final image.

In order to overcome the problem that deep layers may be weakened by shallow layers as the depth increases, we introduce a multi-scale feature extraction (MFE) block in our proposed model. The MFE block increases the field of view by conducting two convolutions of kernel sizes 3 × 3 and 7 × 7, so as to generate 64-deep feature maps which can capture as much information as possible. Then, two residual dilated convolution blocks are used to enlarge the receptive field. The resulting output is merged by the concatenation operation. The architecture of the MFE block is shown in Figure 2b, where RDC denotes the residual dilated convolution function; PR,Cat, and CPR, respectively, denote the functions of PreLU, concatenation operation, and Conv + PreLU; and $C_{3}$ and $C_{7}$, respectively, denote convolutions with kernel sizes of 3 × 3 and 7 × 7. The description of the MFE block can be represented as follows:(4)$$y_{MFE}=C_{3}PR(Cat(RDC(RDC(C_{3}PR(y))),RDC(RDC(C_{7}PR(y)))))$$

#### 3.2.2. Attention-Guided Filter Block

The down-sampling operation of UNet may result in the loss of spatial information. Moreover, this problem cannot be well recovered by skipping connections or up-sampling operations. Therefore, we propose an attention-guided filter block. Specifically, we add a trainable guide filter module after each down-sampling operation to better recover the spatial information loss.

The basic principle of the guided filter is as follows. For input image P and the guided image I, the guided filter is to compute local coefficients $\left(a_{k},b_{k}\right)$, which computes the output image $Q_{i}$ by
(5)$$Q_{i}=a_{k}I_{i}+b_{k},\forall i\in \omega_{k}$$
where $\omega_{k}$ is a local window.

When the input image is the same as the guided image, the guided filter becomes edge-preserving filter. Hence, we choose the case I=P in our model. The specific process is shown in Figure 3. We operate the attention mechanism on the image I to improve the sensitivity of the channel features. The coefficient $a_{k}$ is obtained by performing the AdaptiveAvgPool, Linear + ReLU, and Linear + Sigmoid operations. The coefficient $b_{k}$ is obtained by performing the Cov + GELU and Cov + PReLU operations on the image I.

#### 3.2.3. Residual Denoising Block

The residual denoising block is shown in Figure 4, which is composed of the bottleneck blocks and the feature maps on the basis of ResNet50 and the DenseNet model. We first use the 3 × 3 convolution to reduce the number of the feature maps by half. Then, we use two 3 × 3 convolutions to take all the previous feature maps as inputs. Finally, the 3 × 3 convolution is utilized to aggregate all the previous feature maps with the denoising block input. Finally, the last convolution generates the same number of feature maps as the inputs of the denoising block.

## 4. Results and Discussion

### 4.1. Experimental Setup

In our experiments, we use the DIV2K dataset [44] and the AID dataset [45] to verify the performance of MAGUNet, respectively. The DIV2K dataset consists of 800 images for training, 100 images for validation, and 100 images for testing. The AID dataset consists of 8000 images for training, 1000 images for validation, and 1000 images for testing. Our model was implemented in Python 3.8 on the basis of PyTorch framework. 

To train our proposed model, we split the original training dataset into input and output blocks of size 64 × 64. We trained our model for color images and gray images, respectively. When training our model for grayscale images, we first convert color images to grayscale images, then add Gaussian white noise to clean image block $x_{i}$, in order to generate noisy image block $y_{i}$. The noise intensity for the training set is in the range σ∈[5,50]. In addition, we apply augmentation techniques, including random vertical, horizontal flips, and 90° rotation, in order to extend the dataset.

We employ Adam optimizer to optimize the network parameters. The regularization parameter appearing in the problem (3) is $\lambda =10^{-5}$ and the initial learning rate is $\alpha_{0}=1.2\times 10^{-4}$, which is halved every two iterations throughout the training of the dataset until its value is $\alpha_{f}=10^{-6}$. The MAGUNet model was trained with a batch size of 16 for 14 epochs.

We first report the loss curves and PSNR curves during training. The results show that MAGUNet is well trained and achieves good denoising results on the validation data. It is worth noting that, as shown in Figure 5, here we train for a certain range of noise rather than a specific noise level, and the weight σ=50 of the noise level increases during training, while the proportion of the noise level remains the same during validation, so it looks like the training results are lower than the validation results.

We first report the Loss curves and PSNR curves during training, and the results show that MAGUNet is well trained and achieves good denoising results on the validation data. It is worth noting that, as shown in Figure 5, here we trained on a certain range of noise rather than a specific noise level, and the weight of 50 noise level was added in the training, but the proportion of 10, 30, and 50 noise levels is the same in the validation, so it seems that the training results are lower than the validation results.

We use Set12 [7], BSD68 [46], Kodak24 [47] datasets for the evaluation of grayscale images, and Set5 [48], LIVE1 [49], McMaster [50] datasets for the evaluation of color images. We compared the MAGUNet model and the enhanced model MAGUNet+ with BM3D [2], IRCNN [9], DnCNN [7], FFDNet [20], ADNet [12], RDUNet [29], and MSANet [33]. The experiment results of all the comparative models are obtained using the respective pre-trained models and the source code tests of the corresponding authors. We select PSNR and SSIM indexes to measure the image denoising effect of different algorithms. We present the results for the noise levels of variance σ=10,30,50. The best PSNR and SSIM results for each noise level are highlighted in red, and the second-best results are highlighted in blue.

### 4.2. Grayscale Common Image Denoising

As shown in Table 1, our model achieves satisfactory results. Specifically, our proposed model achieved the best results in the Set12 dataset, outperforming the BM3D algorithm by an average of 1.29 dB, outperforming DnCNN, IRCNN, FFDNet, and ADNet by more than 0.4 dB, and outperforming RDUNet and MSAUNet by an average of 0.04 dB. Additionally, our model has significant advantages over the BSD68 dataset and the Kodak24 dataset. Moreover, the results obtained in the training noise level range of σ∈[5,50] are slightly higher than those obtained by MSAUNet’s separate training of each noise level.

The visual denoising effect of different methods is shown in Figure 6, Figure 7, Figure 8 and Figure 9. The image denoising results of BSD68 dataset 3096 are shown in Figure 6. Our model can recover more detail of the tail letter and reduce artifacts in the letter A. From Figure 7, we can see that the BM3D, DnCNN, IRCNN, FFDNet, ADNet algorithms form a large number of artifacts, and for RDUNet, MSAUNet, our method performs well in detail preservation and smoothing. Furthermore, from Figure 8 and Figure 9, our model outperforms the other models on the tablecloth texture, clearly recovering the detailed texture without excessive smoothing.

### 4.3. Color Common Image Denoising

Table 2 presents the denoising results of color images with different methods on Set5, LIVE1, and McMaster datasets with Gaussian white noise of variances of 10, 30, and 50. We can see that our model also greatly outperforms other competing methods on color images, having a slightly higher effect than MSAUNet. It should be noticed that MSAUNet is trained separately at each noise level. From Figure 10, Figure 11, Figure 12 and Figure 13, we compared our model with the visual denoising effects of CBM3D, DnCNN, IRCNN, FFDNet, ADNet, RDUNet, and MSAUNet. From Figure 10, our method has no artifacts on the lamp post, unlike other methods. Consequently, our model has an advantage in detail retention. As shown in Figure 11, the image obtained by our method is smoother in the uniform region and the edge region compared with other methods. Additionally, it can be seen from Figure 12 and Figure 13 that our method is richer in color and details.

### 4.4. Remote Sensing Image Denoising

Table 3 lists the color image denoising results of different methods on selected test datasets with noise levels of 10, 30, and 50. It can be seen that our model outperforms the other competing methods. Figure 14 and Figure 15 show the denoising results of different methods for images with a noise level of 50. Our model is compared with BM3D, DnCNN, IRCNN, FFDNet, ADNet, and RDUNet for image denoising. Figure 14 shows that the denoised image obtained by the proposed method is smoother in the uniform region and sharper in the edge region, as compared to other methods. In Figure 15, the image handled by our method has richer and clearer color and detail content compared to the state-of-the-art denoising methods DnCNN, ADNet, and BM3D.

### 4.5. Ablation Study

To further verify the effectiveness of our proposed model, we conducted the following ablation experiments to show the effects of the denoising block, guided filter layer, as well as the MFE block. Case 0 denotes our proposed model. Case 1 represents our MAGUNet without the MFE block. Case 2 represents MAGUNet without the attention-guided block. Case 3 indicates MAGUNet model with only RDB blocks. Case 4 represents adding a normal guided filter for training to the MAGUNet model with RDB blocks. Table 4 shows the denoising results for different cases on the McMaster dataset with a Gaussian noise of the variance 50.

For the effectiveness of multi-scale feature extraction blocks, the results about Case 0, 2, and 3 show that the multi-scale features extracted by adding extra MFE blocks can improve the network performance.

For the effectiveness of attention-guided filter blocks, by comparing Cases 0, 1, and 3, the effectiveness of the attention-guided filter can be observed. In the absence of attention guidance in the denoising network, learning the global average directly yields that Case 3 cannot fully focus on structural information during the denoising process. Attention guidance can supplement the down-sampled feature information and provide guidance for the extraction of structural information. In addition, Cases 3 and 4 have similar network structures, but Case 4 adds trainable guided filters. Experimental results show that the performance of attention-guided filter blocks is better than that of traditional guided filters.

Note that as the model complexity increases, so does its computational cost and performance.

## 5. Discussion

Deep learning-based image denoising methods are becoming increasingly popular among researchers due to their ease of implementation and fast processing. In this paper, we analyzed the limitations of down-sampling in U-shaped networks and propose an attention-directed filtering to overcome these limitations. We conducted comparative experiments between the proposed method and other methods. The experiment results showed that our model significantly improves the denoising performance. Also, we foresee the potential of combining traditional denoising techniques with deep learning models for more effective noise reduction strategies.

The MFE and AGF blocks in MAGUNet are designed to address the limitations of the down-sampling problem. Specifically, the MFE block is trained to increase the receptive domain and connect different features for extracting more image information, while the AGF block can preserve the information of image edges after each down-sampling operation, relying on its edge retention capabilities. We performed the corresponding ablation experiments to elucidate the function of the MFE and AGF blocks. Indeed, removing either MFE or AGF leads to the reduction of PSNR and SSIM, which proves the importance of these two blocks in the process of image denoising.

However, our work still has some limitations. It lacks the disadvantage of real-time applications due to its high computational complexity. In this paper, we have only used Gaussian noise to train the dataset and failed to train on various types of noise. And failing to be tested and evaluated in the real world, the proposed model may only be generally applicable to certain image denoising tasks, which may require additional optimization for specific use cases. In addition, while the model performs well in terms of PSNR and SSIM, there are other metrics that may help optimize and evaluate the model. These issues will also be further explored in future research.

## 6. Conclusions

In this paper, we propose a residual UNet model that introduces an attention-guided filter and multi-scale feature extraction. Instead of using a standard input block, we use a multi-scale feature extraction block as the input block. Our MFE blocks placed in the shallow layer of the network are designed to increase the acceptance domain and connect different features. In addition, we develop the attention-guided filter to keep the edge, which has good detail retention ability after each down-sampling operation. We use a global residual network strategy to model residual noise, which does not require the information about the noise level in the noisy image. Experiment results show that our proposed method is competitive with the state-of-the-art methods.

## Figures and Tables

**Figure 1 sensors-23-07044-f001:**
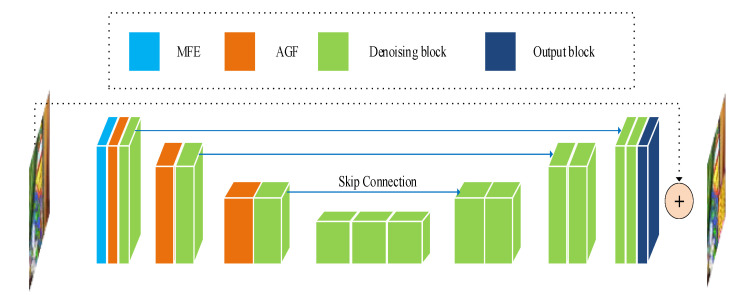
The structures of the encoder and decoder are the main body of the proposed MAGUNet, which contains four main blocks, namely, MFE block, AGF block, denoising block, and output block. The encoder and decoder parts are connected by a multi-scale feature extraction block, attention-guided filter blocks, and residual modules. The main task of the encoder part is to extract the low-level features of the image. The decoder part is responsible for recovering the high-level features of the image while removing the noise.

**Figure 2 sensors-23-07044-f002:**
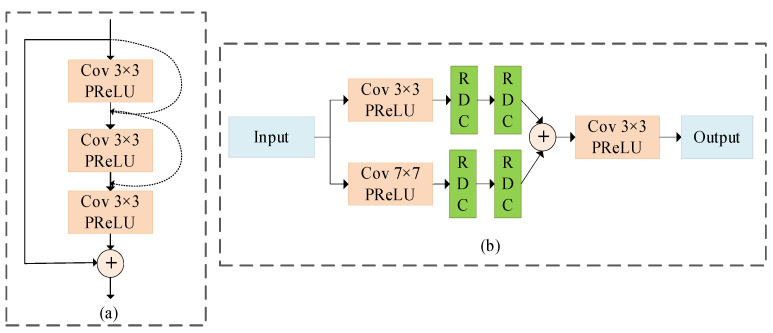
Details of the multi-scale feature extraction block. (**a**) Residual dilated convolution block (RDC), (**b**) multi-scale feature extraction block structure. The practical application of different convolutional kernels is to extend the receptive field for multi-scale feature extraction.

**Figure 3 sensors-23-07044-f003:**
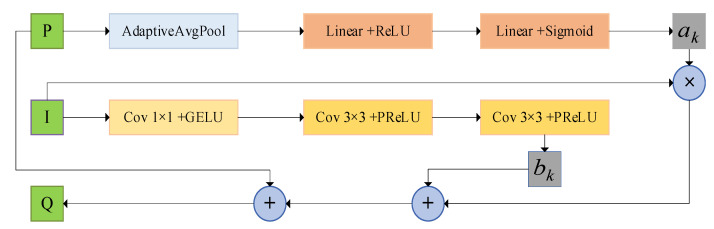
Attention-guided filter block. The coefficients $a_{k}$ are computed by performing the channel attention mechanism on the input image P, and the coefficient $b_{k}$ is computed by convolving the guided image I. Separately, the output image Q is obtained from the residual structure.

**Figure 4 sensors-23-07044-f004:**
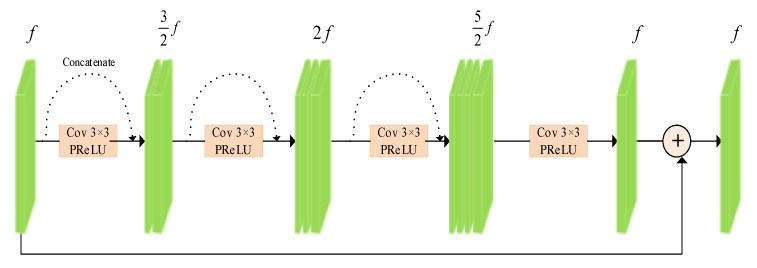
Residual denoising block. Reuse of feature maps by using densely connected denoising blocks.

**Figure 5 sensors-23-07044-f005:**
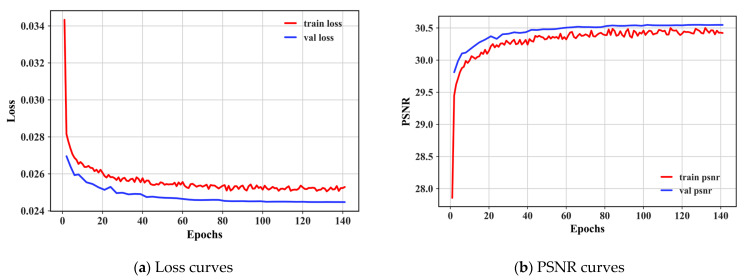
Loss and PSNR curves for training MAGUNet in AWGN denoising. The training dataset is the DIV2K training set, and the PSNR results are computed on the DIV2K dataset at a noise level σ∈[5,50].

**Figure 6 sensors-23-07044-f006:**
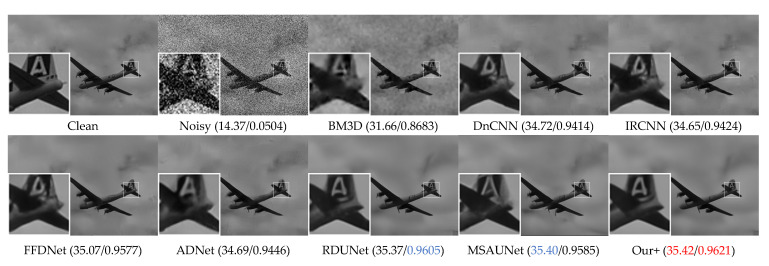
Comparison of the visual quality of different algorithms for a 3096 image from the BSD68 dataset with Gaussian noise of variance 50.

**Figure 7 sensors-23-07044-f007:**
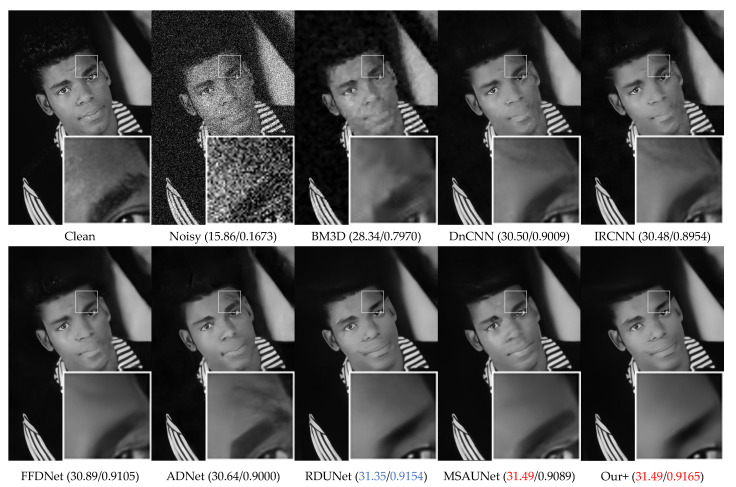
Comparison of the visual quality of different algorithms for a 302008 image from the BSD68 dataset with Gaussian noise of variance 50.

**Figure 8 sensors-23-07044-f008:**
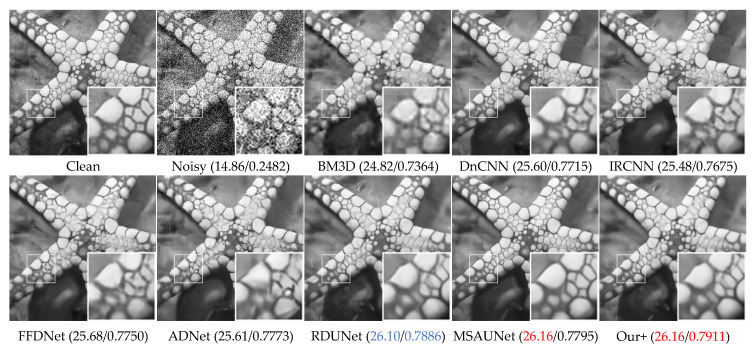
Comparison of the visual quality of different algorithms for a starfish image from the Set12 dataset with Gaussian noise of variance 50.

**Figure 9 sensors-23-07044-f009:**
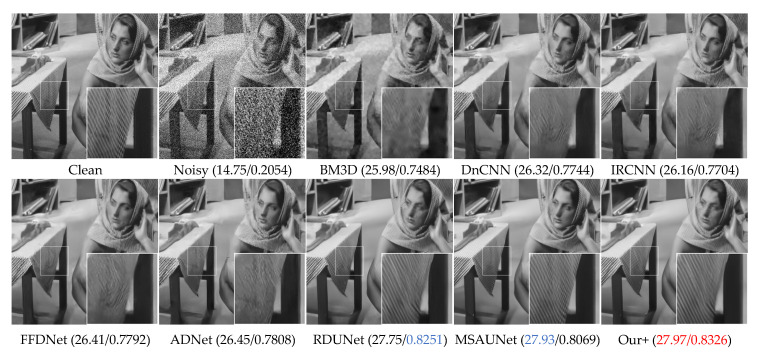
Comparison of the visual quality of different algorithms for a Barbara image from the Set12 dataset with Gaussian noise of variance 50.

**Figure 10 sensors-23-07044-f010:**
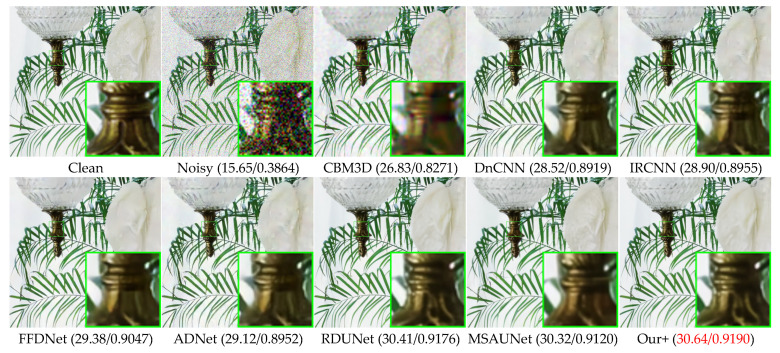
Comparison of the visual quality of different algorithms for 4 images from the McMaster dataset with a noise level of 50.

**Figure 11 sensors-23-07044-f011:**
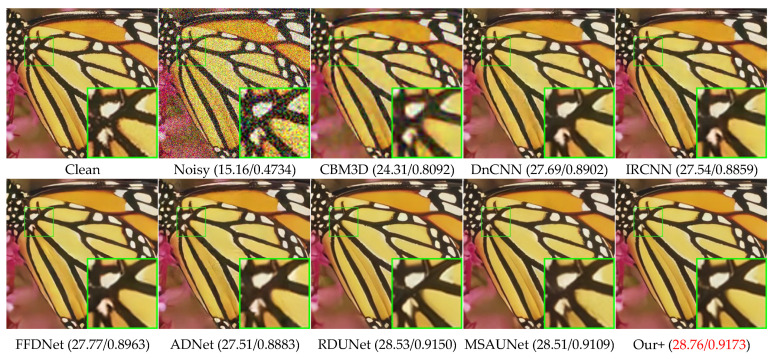
Comparison of the visual quality of different algorithms for butterfly images from the Set5 dataset with a noise level of 50.

**Figure 12 sensors-23-07044-f012:**
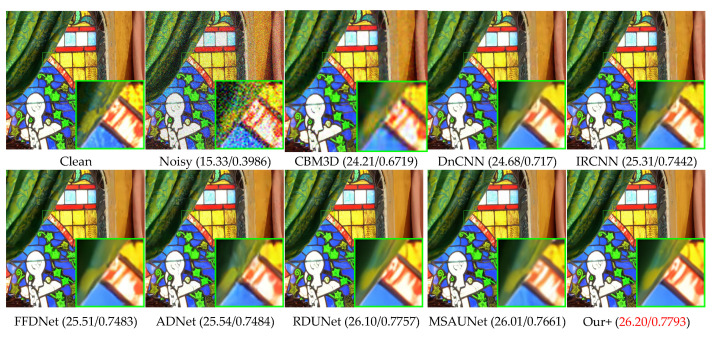
Comparison of the visual quality of different algorithms for 1 image from the McMaster dataset with a noise level of 50.

**Figure 13 sensors-23-07044-f013:**
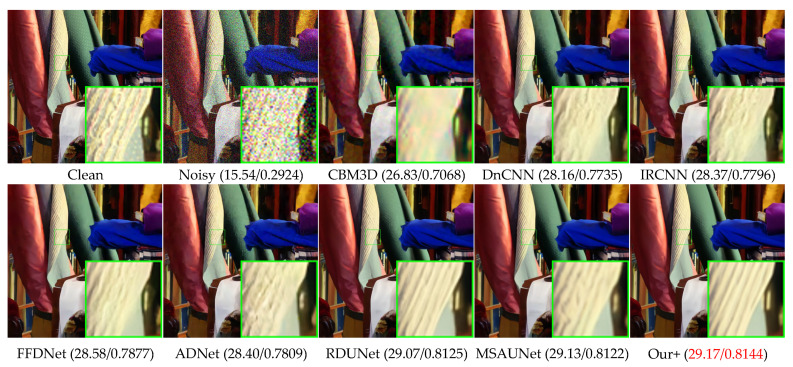
Comparison of the visual quality of different algorithms for 2 images from the McMaster dataset with a noise level of 50.

**Figure 14 sensors-23-07044-f014:**
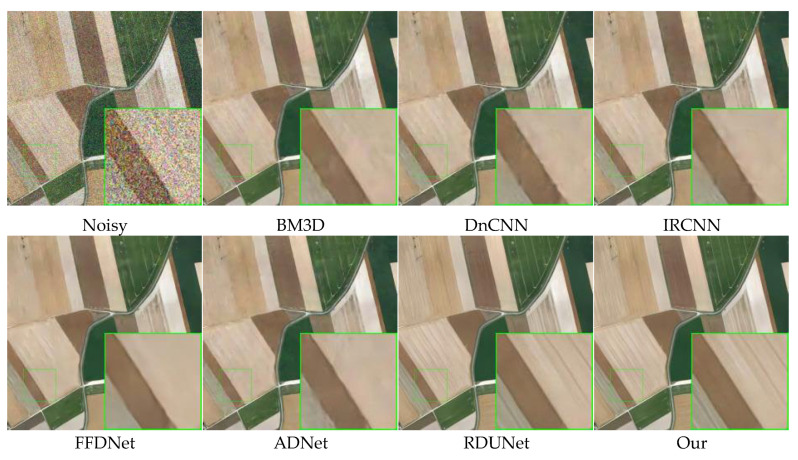
Comparison of the visual quality of different algorithms for a farmland image from the testing dataset with a noise level of 50.

**Figure 15 sensors-23-07044-f015:**
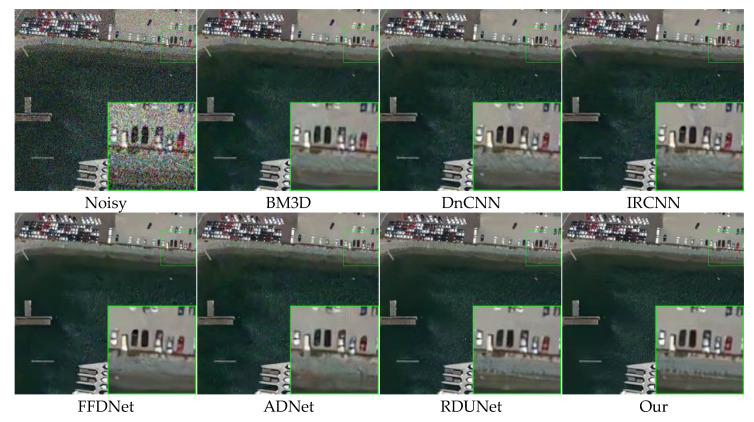
Comparison of the visual quality of different algorithms for a harbor image from the testing dataset with a noise level of 50.

**Table 1 sensors-23-07044-t001:** PSNR (dB) and SSIM results of different denoising methods on Set12, BSD86 and Kodak24 grayscale datasets with different noise levels.

Method\Gray	Set12						BSD68						Kodak24					
	10.00		30.00		50.00		10.00		30.00		50.00		10.00		30.00		50.00	
	PSNR	SSIM	PSNR	SSIM	PSNR	SSIM	PSNR	SSIM	PSNR	SSIM	PSNR	SSIM	PSNR	SSIM	PSNR	SSIM	PSNR	SSIM
BM3D	34.08	0.9204	28.69	0.8199	26.18	0.7411	33.16	0.9160	27.59	0.7671	25.41	0.6722	34.07	0.9113	28.68	0.7779	26.42	0.6918
DnCNN	34.52	0.9241	29.52	0.8422	27.18	0.7816	33.73	0.9241	28.35	0.7982	26.23	0.7164	34.68	0.9207	29.52	0.8082	27.39	0.7364
IRCNN	34.71	0.9272	29.45	0.8393	27.12	0.7804	33.75	0.9263	28.27	0.7993	26.19	0.7169	34.67	0.9212	29.42	0.8064	27.33	0.7354
FFDnet	34.64	0.9270	29.60	0.8464	27.30	0.7899	33.77	0.9266	28.39	0.8031	26.29	0.7239	34.72	0.9223	29.58	0.8122	27.49	0.7434
ADNet	34.63	0.9247	29.62	0.8449	27.29	0.7874	33.65	0.9216	28.32	0.7949	26.22	0.7148	34.67	0.9200	29.51	0.8066	27.4	0.7367
RDUNet	34.99	0.9315	29.96	0.8552	27.72	0.8044	33.97	0.9297	28.58	0.8099	26.48	0.7346	35.00	0.9262	29.86	0.8228	27.78	0.7577
MSANet	\	\	30.00	0.8366	27.72	0.7864	\	\	28.62	0.7939	26.52	0.7229	\	\	29.91	0.8112	27.82	0.7516
Ours	35.03	0.9320	29.96	0.8548	27.70	0.8044	33.99	0.9298	28.56	0.8081	26.45	0.7318	35.04	0.9263	29.86	0.8222	27.75	0.7559
Ours+	35.07	0.9324	30.01	0.8556	27.76	0.8057	34.02	0.9301	28.59	0.8090	26.48	0.7329	35.08	0.9267	29.91	0.8233	27.81	0.7574

**Table 2 sensors-23-07044-t002:** PSNR (dB) and SSIM results of different denoising methods on Set5, LIVE1, and McMaster color datasets with different noise levels.

Method\Color	Set5						LIVE1						McMaster					
	10		30		50		10		30		50		10		30		50	
	PSNR	SSIM	PSNR	SSIM	PSNR	SSIM	PSNR	SSIM	PSNR	SSIM	PSNR	SSIM	PSNR	SSIM	PSNR	SSIM	PSNR	SSIM
BM3D	36.02	0.9392	30.93	0.8592	28.69	0.8092	35.82	0.9484	30.08	0.8542	27.66	0.7816	35.91	0.9336	30.84	0.8512	28.54	0.79
DnCNN	35.74	0.9321	31.15	0.864	28.96	0.8146	35.69	0.9485	30.35	0.864	27.95	0.7951	34.79	0.9226	30.79	0.854	28.62	0.7986
IRCNN	36.13	0.9392	31.17	0.8655	29.00	0.8172	36.00	0.9497	30.36	0.8648	27.97	0.7979	36.45	0.9406	31.31	0.8642	28.93	0.8069
FFDNet	36.16	0.9397	31.35	0.8689	29.24	0.8252	36.07	0.9508	30.49	0.8663	28.10	0.7988	36.45	0.9414	31.53	0.8701	29.19	0.8149
ADNet	35.97	0.9355	31.21	0.8664	28.99	0.8158	35.97	0.9501	30.37	0.8639	27.93	0.792	36.27	0.939	31.33	0.8658	29.03	0.936
RDUNet	36.54	0.9422	31.83	0.8797	29.69	0.8398	36.51	0.9546	31.00	0.8789	28.64	0.8195	36.95	0.9469	32.09	0.885	29.79	0.8378
MSANet	\	\	31.83	0.8865	29.69	0.8437	\	\	30.96	0.8816	28.64	0.8224	\	\	32.10	0.8884	29.82	0.8409
Ours	36.57	0.9426	31.84	0.8786	29.72	0.8387	36.54	0.9546	31.01	0.8782	28.64	0.8181	37.05	0.9477	32.13	0.8851	29.82	0.8373
Ours+	36.61	0.9429	31.89	0.8795	29.76	0.8396	36.58	0.9549	31.06	0.8790	28.70	0.8192	37.11	0.9483	32.20	0.8864	29.90	0.8393

**Table 3 sensors-23-07044-t003:** PSNR (dB) and SSIM results of different denoising methods on testing datasets with different noise levels.

Method	10		30		50	
	PSNR	SSIM	PSNR	SSIM	PSNR	SSIM
BM3D	36.81	0.9416	31.51	0.8349	29.30	0.7579
DnCNN	36.71	0.9401	31.49	0.8336	29.32	0.7588
IRCNN	36.75	0.9408	31.43	0.8333	29.31	0.7607
FFDNet	36.81	0.9415	31.58	0.8348	29.43	0.7600
ADNet	36.76	0.9408	31.49	0.8325	29.31	0.7553
RDUNet	37.06	0.9446	31.96	0.8482	29.87	0.7814
Ours	37.25	0.9462	32.04	0.8500	29.90	0.7822
Ours+	37.28	0.9466	32.07	0.8509	29.95	0.7838

**Table 4 sensors-23-07044-t004:** Ablation investigation for MAGUNet. Average PSNR (dB) and SSIM values on McMaster for a noise of level 50.

	Case 0	Case 1	Case 2	Case 3	Case 4
MFE	√	×	√	×	×
RDB	√	√	√	√	√
AGF	√	√	×	×	×
PSNR	29.8235	29.8107	29.7970	29.7907	29.7895
SSIM	0.8373	0.8360	0.8347	0.8328	0.8327
Complexity	2.61 GMac	2.26 GMac	2.55 GMac	2.20 GMac	2.48 GMac

## Data Availability

Datasets used in this paper are open access and are available from: DIV2K is openly available in “NTIRE 2017 challenge on single image super-resolution: Dataset and study”, reference number [45]; Set12 is openly available in “Beyond a Gaussian Denoiser: Residual Learning of Deep CNN for Image Denoising”, reference number [8]; BSD68 is openly available in “Fields of Experts: a framework for learning image priors”, reference number [46]; Kodak24 is openly available in “Kodak lossless true color image suite: PhotoCD PCD0992” at url: http://r0k.us/graphics/kodak.182(accessed on 14 July 2023), reference number [47]. Set5 is openly available in “Accurate Image Super-Resolution Using Very Deep Convolutional Networks”, reference number [48]; LIVE1 is openly available in “A Statistical Evaluation of Recent Full Reference Image Quality Assessment Algorithms”, reference number [49]; McMaster is openly available in “Color demosaicking by local directional interpolation and nonlocal adaptive thresholding”, reference number [50].

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
