# Peer review of "A Residual UNet Denoising Network Based on Multi-Scale Feature Extraction and Attention-Guided Filter"

_sensors, 2023, doi:10.3390/s23167044_

Round 1

Reviewer 1 Report

In general, the submitted manuscript is well written and well structured. It is easy to navigate within the manuscript. The main contributions are clearly stated from Line 61 to Line 80. In Line 66, the authors write that "multi-scale feature extraction block to extract more useful features from the input noise image". The authors should mention here that the usefulness of multi-scale feature have been pointed out in other fields of computer vision, such as image quality (No-Reference Image Quality Assessment with Multi-Scale Orderless Pooling of Deep Features, 2021), visual saliency (Visual saliency based on multiscale deep features, 2015), or re-identification (Multi-scale deep learning architectures for person re-identification, 2017).

I think that the description of the proposed method is good and suitable for a scientific publication. The figure captions should be self-contained. Now they are very short and not too informative. 

The presentation of the experimental results is good. The proposed method is compared to several other methods. However, the performance difference in several cases seems rather small. Could you that the performance gain is significant? Further, it would be good to depict the distribution of these performance metrics (PSNR, SSIM) on the test set. Since deep learning involves a lot of experiments, the publication of training curves would be nice. How does the performance of the proposed method depend on the ratio of the training data?

Reviewer 2 Report

The authors have developed a new denoising network called the residual UNet. They added attention-guided filters and multi-scale feature extraction blocks to make the network better at removing noise from images while keeping essential details. The new model performed well compared to advanced models, effectively reducing noise and making the images sharper. The overall idea sounds interesting, the main highlighted points follow such as:

1) It is essential to highlight that the proposed model may only be universally applicable to some image-denoising tasks, potentially requiring additional optimization for specific use cases. Conversely, the significant omission of a detailed analysis regarding the computational complexity of the proposed model, as presented in the paper, emerges as a crucial critique, possibly posing a limitation for specific applications.

2) While the paper outlines an interesting approach in the form of a residual UNet denoising network, MAGUNet, and its implementation, there are several aspects of the methodology that could be enhanced:

a) Dataset Expansion: The proposed model was solely trained on the DIV2K dataset. To increase the model's generalizability, it could be beneficial to use additional diverse datasets encompassing a variety of image noise types and image contexts.

b) Cross-platform testing: The model was developed using the PyTorch framework. For broader usability and to ensure the model's effectiveness across different platforms, it may be helpful to evaluate its performance with other deep learning frameworks, such as TensorFlow or Keras.

c) Performance Benchmarking: While the model was compared against several state-of-the-art models, providing a more thorough comparison would be advantageous, including lesser-known or older models. This comparison could provide a more comprehensive performance landscape, highlighting the model's improvements and still-existing limitations.

d) Computational Complexity Analysis: The paper needs a detailed examination of the computational complexity of the proposed model. This analysis is crucial as it would provide insights into the model's efficiency, a significant aspect, especially for real-time applications.

e) Robustness Testing: Lastly, the model should be subjected to various types of noise, including worst-case scenarios. This robustness testing would evaluate the model's performance in more challenging situations and provide valuable feedback for potential model optimization.

3) While the experimental results are commendable, there remain areas for potential enhancements to improve upon the presented work:

a) Test on Multiple Datasets: Though the model achieved significant scores on the DIV2K dataset, testing it on other image datasets could further substantiate its robustness and generalization capabilities. Diverse datasets would help the model to generalize to a broader range of noise types and image contexts.

b) Real-world Evaluation: While benchmarks like PSNR and SSIM are valuable, they might only sometimes correlate with human perception of image quality. Therefore, it would be beneficial to include real-world image-denoising tasks in the evaluation or user studies to validate the effectiveness of the proposed model from a human visual perception perspective.

c) Optimize for Other Metrics: Although the model performed well in terms of PSNR and SSIM, it might be helpful to optimize and evaluate the model for other metrics as well, like Mean Squared Error (MSE), or perceptual metrics such as Multi-Scale Structural Similarity (MS-SSIM), or Learned Perceptual Image Patch Similarity (LPIPS).

d) Consider Lower-resource Scenarios: The paper could provide insights into how the model performs under constraints, such as reduced computational resources or limited training data. Providing this information would allow potential users to understand the model's practicality in various situations.

e) Model Interpretability: While the model's performance is essential, it might also be beneficial to include an analysis of why the model works, which areas it excels in, and its shortcomings. This could involve visualizing the filters and attention maps or examining the impact of different components of the network architecture on the overall performance.

f) Discussion on Failure Cases: Finally, analyzing cases where the model performs poorly would provide valuable insights for further research. Understanding these failure modes can guide future work to make the model more robust.

3.1) If implementation of the improvements above may not be feasible due to constraints such as time or resources, it would be highly beneficial to incorporate a detailed discussion section in the paper.

This section could not only elucidate the experimental results more comprehensively but also highlight the aspects of the methodology that could be improved upon or explored further.

a) Explaining the Results: Begin by interpreting the results so that even those without a deep understanding of the topic can comprehend. This includes simplifying complex ideas or using illustrative examples to highlight key findings.

b) Implications and Future Work: Discuss the implications of your findings for the broader field of image denoising. Consider the impact of the results on both positive and negative real-world applications. Following this, outline potential avenues for future research, guided by the limitations of the current study and the results obtained.

c) Model Limitations: Clearly outline the limitations of the MAGUNet model. Discuss where it excels, where it falls short, and why this might be the case. This honesty increases the credibility of the paper and serves as a guide for future researchers in the field.

d) Possible Improvements: Here, the points for enhancements suggested earlier could be revisited. These can be proposed as future directions that subsequent researchers could explore. Discuss why these improvements would be significant and how they could impact the field.

e) Connections to Other Works: Discuss how this work relates to previous research in the field. How does it build upon what is already known? How does it diverge from or challenge existing knowledge or methodologies? Adding this detailed discussion section will not only help to contextualize the results of the current work fully. However, it will also provide an invaluable resource for other researchers who wish to build upon this work in the future.

4) In Section 2 of the paper, the authors undertake a comprehensive review of numerous leading-edge denoising models, including but not limited to DnCNN, FFDNet, and RIDNet. It meticulously dissects the deficiencies of these contemporary models, underscoring the necessity for a more efficient denoising model capable of superior noise reduction while preserving image specifics. Moreover, the authors delve into the significance of multi-scale feature extraction and attention-guided filters as vital components of denoising models. They propose that leveraging these techniques could enhance the overall model's efficacy. The literature review section provides an exhaustive overview of the present-day pinnacle of denoising models, setting the stage for the need to delve deeper into this field. In addition, it would be compelling to consider integrating more advanced techniques, such as generative models or reinforcement learning approaches, to tackle the denoising problem. Further research could also investigate the possibility of an adaptive model capable of adjusting its denoising strength based on the type and level of noise present in the image. Future studies could also delve into the potential of combining traditional denoising techniques with deep learning models for more effective noise reduction strategies.

Final Review Remarks

Upon a comprehensive review of the manuscript presented and considering the points mentioned above, a significant revision is necessary to realize and articulate this research's potential fully.

While the authors' development of a novel residual UNet denoising network, the MAGUNet, is both exciting and potentially significant, several areas of the methodology and analysis could be further refined and expanded to increase the overall impact of the study.

Some of the critical areas to be addressed include the broad applicability of the proposed model, the inclusion of detailed computational complexity analysis, dataset expansion, cross-platform testing, a more exhaustive performance benchmarking, robustness testing, and several other aspects related to the optimization, evaluation, and interpretation of the model's performance. Additionally, considerations for real-world applications, alternative metrics, model constraints, and failure case discussions should be more thoroughly explored.

Moreover, if the implementation of the above improvements is not feasible due to certain constraints, a detailed discussion section in the paper would be highly beneficial. This section could comprehensively elucidate the experimental results and highlight aspects of the methodology that could be improved or explored further.

Lastly, while the authors have provided a thorough review of numerous leading-edge denoising models in Section 2, suggesting the need for further research in this field, the integration of more advanced techniques and the potential of combining traditional denoising techniques with deep learning models for more effective noise reduction strategies could also be explored.

In conclusion, the necessity for a significant revision is suggested, with the belief that addressing these points will significantly strengthen the paper, thus making a more substantial contribution to the field of image denoising. The authors are encouraged to address these points thoroughly in their revisions. This decision should encourage the authors, as their work's novelty and potential impact are apparent. These comments are meant to guide the enhancement of the manuscript to ensure that it meets the highest standards of scientific research.

Round 2

Reviewer 1 Report

The authors modified the manuscript with respect to the reviewer's concerns. The manuscript can be published now.

Reviewer 2 Report

The authors discussed and debated appropriately all suggestions and points previously mentioned in the review round. Despite considering there are a lot of points of improvement in the paper, and also understanding that much of them can be conducted in future works. The reviewer recognizes the critical gaps were clarified, and because of this, the recommendation is to approve the paper in the present form.